# Evaluation of Spectrum-Aided Visual Enhancer (SAVE) in Esophageal Cancer Detection Using YOLO Frameworks

**DOI:** 10.3390/diagnostics14111129

**Published:** 2024-05-29

**Authors:** Chu-Kuang Chou, Riya Karmakar, Yu-Ming Tsao, Lim Wei Jie, Arvind Mukundan, Chien-Wei Huang, Tsung-Hsien Chen, Chau-Yuan Ko, Hsiang-Chen Wang

**Affiliations:** 1Division of Gastroenterology and Hepatology, Department of Internal Medicine, Ditmanson Medical Foundation Chia-Yi Christian Hospital, Chia-Yi 60002, Taiwan; vacinu@gmail.com; 2Obesity Center, Ditmanson Medical Foundation Chia-Yi Christian Hospital, Chia-Yi 60002, Taiwan; 3Department of Medical Quality, Ditmanson Medical Foundation Chia-Yi Christian Hospital, Chia-Yi 60002, Taiwan; 4Department of Mechanical Engineering, National Chung Cheng University, Chia-Yi 62102, Taiwan; karmakarriya345@gmail.com (R.K.); d09420002@ccu.edu.tw (Y.-M.T.); d09420003@ccu.edu.tw (A.M.); 5Department of Computer Science, Multimedia University (Cyberjaya), Persiaran Multimedia, Cyberjaya 63100, Malaysia; jaylim.wjie@gmail.com; 6Department of Gastroenterology, Kaohsiung Armed Forces General Hospital, 2, Zhongzheng 1st. Rd., Lingya District, Kaohsiung City 80284, Taiwan; forevershiningfy@yahoo.com.tw; 7Department of Nursing, Tajen University, 20, Weixin Rd., Yanpu Township 90741, Pingtung County, Taiwan; 8Department of Internal Medicine, Ditmanson Medical Foundation Chia-Yi Christian Hospital, Chia-Yi 60002, Taiwan; cych13794@gmail.com; 9Department of Medical Research, Dalin Tzu Chi Hospital, Buddhist Tzu Chi Medical Foundation, No. 2, Minsheng Road, Dalin, Chia-Yi 62247, Taiwan; 10Director of Technology Development, Hitspectra Intelligent Technology Co., Ltd., Kaohsiung City 80661, Taiwan

**Keywords:** esophageal cancer, hyperspectral imaging, SAVE, dysplasia, YOLOv5, YOLOv8, narrowband imaging, white-light imaging

## Abstract

The early detection of esophageal cancer presents a substantial difficulty, which contributes to its status as a primary cause of cancer-related fatalities. This study used You Only Look Once (YOLO) frameworks, specifically YOLOv5 and YOLOv8, to predict and detect early-stage EC by using a dataset sourced from the Division of Gastroenterology and Hepatology, Ditmanson Medical Foundation, Chia-Yi Christian Hospital. The dataset comprised 2741 white-light images (WLI) and 2741 hyperspectral narrowband images (HSI-NBI). They were divided into 60% training, 20% validation, and 20% test sets to facilitate robust detection. The images were produced using a conversion method called the spectrum-aided vision enhancer (SAVE). This algorithm can transform a WLI into an NBI without requiring a spectrometer or spectral head. The main goal was to identify dysplasia and squamous cell carcinoma (SCC). The model’s performance was evaluated using five essential metrics: precision, recall, F1-score, mAP, and the confusion matrix. The experimental results demonstrated that the HSI model exhibited improved learning capabilities for SCC characteristics compared with the original RGB images. Within the YOLO framework, YOLOv5 outperformed YOLOv8, indicating that YOLOv5’s design possessed superior feature-learning skills. The YOLOv5 model, when used in conjunction with HSI-NBI, demonstrated the best performance. It achieved a precision rate of 85.1% (CI95: 83.2–87.0%, *p* < 0.01) in diagnosing SCC and an F1-score of 52.5% (CI95: 50.1–54.9%, *p* < 0.01) in detecting dysplasia. The results of these figures were much better than those of YOLOv8. YOLOv8 achieved a precision rate of 81.7% (CI95: 79.6–83.8%, *p* < 0.01) and an F1-score of 49.4% (CI95: 47.0–51.8%, *p* < 0.05). The YOLOv5 model with HSI demonstrated greater performance than other models in multiple scenarios. This difference was statistically significant, suggesting that the YOLOv5 model with HSI significantly improved detection capabilities.

## 1. Introduction

Esophageal cancer (EC) ranks eighth among the most frequently diagnosed cancers and is the sixth leading cause of global cancer-related mortality [1,2]. The prevalence of EC displays substantial variability across countries and populations, primarily stemming from differences in the prevalence of underlying risk factors and the distribution of its subtypes [3]. Despite these discrepancies, EC survival rates remain markedly low, typically ranging from 10% to 30% five years post-diagnosis in most countries [4,5]. Early detection of EC is crucial to enhancing patient survival. However, identifying EC in its early stages proves challenging because of the absence of obvious symptoms. The cancer’s aggressive nature also leads to a poor prognosis, even for experienced endoscopists. This difficulty also contributes to the potential oversight of early symptoms, and by the time EC is detected, it often advances to the second or third stages [6]. Aside from endoscopy, which is essential for determining the extent of EC within the body, radiological techniques serve as a vital supplementary method. Transesophageal ultrasound is a highly effective method of evaluating the extent of tumor invasion and identifying the presence of nearby lymph-node involvement. This provides crucial information for planning appropriate treatment strategies [7]. Computed tomography (CT) scans are commonly used to evaluate the spread of EC and determine the presence of distant metastases, but their ability to detect early-stage tumors and lymph node involvement is limited [8]. Magnetic resonance imaging (MRI) is not as frequently utilized as CT for EC. However, it offers superior contrast resolution, which can be beneficial for precise anatomical evaluations, especially when assessing the advanced stage of tumor infiltration into nearby structures [9]. However, endoscopy assumes a crucial role in identifying cancers and other tissue irregularities. Endoscopic images obtained through conventional mechanisms can undergo alterations owing to tissue secretion or instrument specifications, potentially leading to diagnostic misjudgments. White-light imaging (WLI) utilizes a wide spectrum of visible light for lesion characterization, whereas narrowband imaging (NBI) combined with hyperspectral imaging (HSI) acquires a large number of bands, often spanning a wide range of wavelengths [10]. Accordingly, a detailed spectral profile is produced for each pixel in an image, thereby enabling the analysis of subtle differences in material composition [11]. HSI is a promising medical imaging technique with extensive potential applications in the field of biomedicine, particularly in disease diagnosis and image-guided surgery [12]. HSI quantifies the light that bounces off tissue and provides data on its structure and spectral characteristics [13]. This imaging technique relies on the concept that various tissues possess distinct spectral reflectance responses, manifested as individual spectral fingerprints [14]. HSI utilizes a continuous narrowband dataset to acquire image data from a specific area of interest [15]. HSI-NBI uses specific filters, namely blue (415 nm) and green (540 nm), to intensify contrast for a meticulous lesion analysis [16]. The blue filter accentuates features with a higher ratio, thereby creating darker contrasts. Conversely, the green filter, synchronized with the secondary absorption peak, unveils deeper lesions in cyan. HSI, extensively applied in diverse fields like air pollution detection [17,18], satellite photography [19,20,21], geology [22,23,24], counterfeit detection [25], military [26,27], agriculture [28,29], etc., emerges as a promising technology when integrated with artificial intelligence deep learning for EC spectral data analysis. Tsai et al., [30] used the HSI method in conjunction with single-shot multibox detector algorithms. The findings demonstrate that using the HSI method facilitates the accurate identification of EC stages and the precise marking of locations. The conversion into HSI enhances lesion features, providing optimal input for model training and ensuring robust results and a novel approach. Over the past few years, numerous scholars have conducted research related to deep learning in the context of EC, showcasing its promise as a potential approach to early detection. Collins et al. [31] proposed an HSI-based CAD tool by using SVM, MLP, and 3DCNN. This showed promise in rapidly detecting colorectal and esophagogastric cancer tissue, with notable performance improvements using patient-specific decision thresholds. Fang et al. [32] used semantic segmentation with NBI and HSI to label early-stage EC, utilizing U-Net as the primary neural network and complementing it with ResNet for precise classification and cancer location prediction. Maktabi et al. [33] utilized HSI data and a multilayer perceptron to discern histopathological features in EC, highlighting HSI’s potential with machine learning for advanced tumor diagnosis. Wu et al. [34] proposed a method for early EC identification, combining endoscopy and hyperspectral endoscopic imaging to detect early cancerous lesions, showcasing potential applications in capsule endoscopy and telemedicine. Zhang et al. [35] used a two-stage deep-learning system (DLS) for the automated detection of EC in barium esophagram, involving the development of a selection network and a classification network. Despite the rapid development of AI in recent years, EC continues to have the lowest survival rate because its symptoms are difficult to detect. The prevailing detection trend predominantly uses classification tasks on entire images, lacking precision in pinpointing lesion locations compared with object detection. Utilizing object detection for EC proves advantageous because it allows for the classification of specific lesion locations, enabling endoscopists to verify and confirm the status on a select number of lesions. Qureshi [36] revealed that YOLO excels in various object detection tasks, including lesion detection [37], skin lesion classification [38], chest abnormality detection [39], breast cancer detection [40], and personal protective equipment detection [41]. Meng et al. [42] examined esophageal squamous cell carcinoma (ESCC) detection using a YOLOv5 model with WLI and HSI images, comparing it with manual detection by endoscopists. They found the diagnostic performance of YOLOv5 promising, suggesting its potential as an assistive tool for less experienced endoscopists in upper gastrointestinal endoscopy. The commendable architecture of YOLO consistently delivers impressive results, enabling it to be a widely adopted framework in medical research owing to its ability to process information rapidly. The present study aimed to explore the urgent need for advancements in the early detection of EC, which continues to present significant diagnostic challenges. HSI offers a sophisticated solution by capturing detailed spectral data that conventional imaging methods cannot obtain. Additionally, the research involved the conversion of WLI images into HSI by using spectrum-aided vision enhancer (SAVE) techniques. Furthermore, this study aimed to implement YOLO frameworks to conduct object detection by precisely identifying the location and stage of cancerous lesions within EC images.

## 2. Materials and Methods

### 2.1. Data Processing

The dataset we utilized incorporated a set of 2761 WLI pictures obtained using a conventional endoscope (CV-290, Olympus, Shinjuku, Tokyo, Japan). These images, sourced from the Ditmanson Medical Foundation of Chia-Yi Christian Hospital, were standardized to a size of 640 × 640 pixels during preprocessing to prevent potential issues, such as insufficient computer memory, and to ensure format uniformity. We studied 150 patients, including 50 who were normal, 50 with dysplasia, and 50 with squamous cell carcinoma (SCC). The age distribution ranged from 40 to 70 years old, and the ratio of men to women was 7:3. Image annotation was conducted using one of the most common platforms known as LabelImg software (version 1.8.6). An XML file was generated, and it was then transformed into a text file [43]. This text file served as the input for the YOLO framework models during the training process. Meanwhile, WLI excels in detecting EC, but its sensitivity is limited in identifying dysplasia. Dysplasia often visually resembles SCC, posing challenges for precise identification through WLI. Accordingly, we used HSI conversion techniques to enhance the images, creating a detailed multidimensional dataset that captured spectrum information across various wavelengths, as illustrated in Figure 1. It is important to highlight that the annotated dataset was transformed into HSI-NBI images by using SAVE techniques throughout the study. Accordingly, two datasets were acquired utilizing WLI and HSI images. Figure 2 illustrates WLI images captured by the original endoscope, depicting the natural color of the images. These images typically highlighted the red coloration of blood vessels within the lesions. Additionally, Figure 2a–c provide an overview of normal esophageal images, dysplasia, and SCC in the WLI dataset. Figure 2d–f showcase HSI images that underwent a conversion process, that is, the SAVE technique, resulting in a reddish brown color. This enhancement improved contrast with the background and also facilitated the parallel feeding of features to machine learning models. Figure 2d–f present an illustration of normal esophageal images, dysplasia, and SCC in HSI dataset. As depicted in Figure 2, dysplasia was challenging to detect even upon personal observation. In certain cases, conspicuous dysplasia may manifest as a polyp, adding to the complexity of identification. Conversely, SCC was typically easier to identify; it often presented discernible residue-like substances. The dataset was partitioned into training, testing, and validation sets at a ratio of 70:15:15. The PyTorch deep-learning framework was constructed on a Windows 11 operating system. The program was scripted in Python and executed in Jupyter Notebook, utilizing Python 3.9.18.

### 2.2. Spectrum-Aided Vision Enhancer (SAVE)

The SAVE technique was crucial to the conversion process in constructing the HSI-NBI dataset for this research. It played a pivotal role in converting an RGB image captured with a digital camera into an HSI image, as illustrated in Figure 1 (the overall flowchart of this project; developed by Hitspectra Intelligent Technology Co., Ltd., 8F.-11-1, No. 25, Chenggong 2nd Rd., Qianzhen Dist., Kaohsiung City 806614, Taiwan (R.O.C.)). Before the transformation, establishing the correlation between the RGB image and the spectrometer across various colors was essential. To achieve calibration, the Macbeth color checker (X-Rite Classic) was assigned as the target. It comprised 24 squares with a diverse array of color samples commonly found in nature, encompassing red, green, blue, cyan, magenta, yellow, and six shades of gray. X-Rite has recently gained prominence as a favored option for color calibration. The endoscopy camera primarily captured images with colors corresponding to the X-Rite board, which acted as the target. The 24-color patch image underwent a transformation into the CIE 1931 XYZ color space. Captured in JPEG format within the standard RGB (sRGB) color space, the R, G, and B values (ranging from 0 to 255) in the sRGB color space were initially adjusted to a smaller gamut between 0 and 1. The gamma function was then applied to transition these scaled sRGB values into linearized RGB values. Using a translation matrix, the linearized RGB values were further converted into the CIE 1931 color space, establishing a numerical correlation between the wavelengths in SAVE and the observed natural colors. Figure 3 shows the schematics of SAVE.

To convert the RGB images to the HSI-NBI images, the endoscope and the spectrometer should be calibrated. In the endoscope part, 8bit JPEG images were stored in the endoscope by using the sRGB color space. Before converting the image from sRGB into *XYZ*, the R, G, and B values (0–255) must be lower (0–1). The gamma function converted the sRGB value into a linear RGB value, which the conversion matrix converted into the *XYZ* color gamut standard. To convert reflection spectrum data (380–780 nm, 1 nm) into the *XYZ* color gamut space in the spectrometer, color matching functions and light source spectrum *S(λ)* were needed. Given that both values were proportional, brightness was calculated from the *XYZ* color gamut space *Y* value. After normalizing the brightness value between 0 and 100 to obtain the luminance ratio *k*, the reflection spectrum data were converted into *XYZ*. The variable matrix *V* was calculated by analyzing camera errors such as nonlinear response, dark current, color filter separation, and color shift. Equation (1) shows how regression analysis on *V* yielded the correction coefficient matrix C for camera errors. The average RMSE of *XYZ_Correct_* and *XYZ_Spectrum_* data was 0.5355. After calibration, *XYZ_Correct_* and spectrometer-measured reflection spectrum data of 24 color patches (*R_Spectrum_*) were compared. The conversion matrix *M* was determined by identifying the key principal components of *R_Spectrum_* through PCA and multiple regression analyses, and the results are shown in Equation (2). *V_Color_* was selected for the multivariate regression analysis of *XYZ_Correct_* and Score because it listed all possible *X*, *Y*, and *Z* combinations. Equation (1) yields the transformation matrix *M*, whereas Equation (4) calculates the analog spectrum (*S_Spectrum_*) using *XYZ_Correct_*.
(1)$$\left[C\right]=\left[{XYZ}_{Spectrum}\right]\times pinv(\left[V\right]),$$
(2)$$\left[{XYZ}_{Correct}\right]=\left[C\right]\times [V],$$
(3)$$\left[M\right]=\left[Score\right]\times pinv(\left[V_{Color}\right]),$$
(4)$${\left[S_{Spectrum}\right]}_{380\sim 780nm}=\left[EV\right]\left[M\right][V_{Color}]$$

The reflection spectrum and analog spectrum of 24 color blocks (*S_Spectrum_*) were compared. Each color block’s RMSE averaged 0.0532. The color difference between *S_Spectrum_* and *R_Spectrum_* can also be represented. This process created the VIS-HSI algorithm, which simulated the camera’s RGB reflection spectrum.

### 2.3. YOLOv5 Model

YOLOv5, marking the fifth evolution of the renowned object detection model, stands out for its commitment to delivering real-time performance coupled with exceptional accuracy. Its architectural design revolves around three pivotal components: the Model Backbone, Model Neck, and Model Head [44,45]. The utilization of the CSP-Darknet53 structure as the backbone ensured the efficient extraction of critical features from input images, laying the foundation for subsequent processing. In the Model Neck, the incorporation of SPPF and PAN structures strategically focuses on generating feature pyramids, a crucial element for ensuring the model’s adaptability to varying object scales during detection. Finally, the Model Head spearheads the conclusive detection phase, applying anchor boxes to produce output vectors comprising class probabilities, objectless scores, and bounding boxes. An overview of the YOLOv5 architecture is shown in Appendix A. These architectural choices collectively contribute to the versatility and effectiveness of YOLOv5 in object detection tasks. Despite its significant impact, as of the current timeframe, YOLOv5 lacks an official research paper, relying on unofficial sources for insights into its architecture and capabilities. YOLOv5 also offers variations in sizes, including nano, small, medium, large, and extra-large, with distinctions in the number of layers and parameters (see Appendix A for the image training set and validation set loss functions and convergence of precision, recall, and mean precision of the WLI and the HSI model using YOLOv5). This feature allows practitioners to tailor the model to specific requirements and computational resources [46].

### 2.4. YOLOv8 Model

Released by Ultralytics in January 2023, YOLOv8 represents the latest version of the YOLO real-time object detection models, providing insightful improvements over its predecessors [47]. By adopting an anchor-free design, YOLOv8 optimizes efficiency by reducing box predictions and expediting the non-maximum suppression process, which is particularly advantageous for handling objects with diverse shapes and aspect ratios [48]. The backbone of YOLOv8 shares similarities with YOLOv5, incorporating the C2f module inspired by the ELAN idea from YOLOv7. This enhances gradient-flow information while maintaining a lightweight structure. The SPPF module is retained at the backbone’s end, ensuring accuracy across various scales. In the neck section, YOLOv8 utilizes the PAN-FPN feature-fusion method and incorporates multiple C2f modules, along with a decoupled head structure inspired by YOLOx, achieving a new level of accuracy by combining confidence and regression boxes. YOLOv8’s flexibility allows seamless support for all YOLO versions and easy switching between them, enabling its compatibility with various hardware platforms [49]. The network architecture diagram, as depicted in Appendix A, showcases that CBS comprises convolution, batch normalization, and SiLu activation functions (see Appendix A for the image training set and validation set loss functions and convergence of precision, recall, and mean precision of the WLI and the HSI model using YOLOv8).

## 3. Results

The training and validation of the models were performed for 500 iterations by using a batch size of 32 and an initial learning rate of 0.01. The model input was standardized to an image size of 640 × 640 pixels. Pre-training augmentation, a crucial component of the YOLO framework, involved performing flips (horizontally and vertically) and translations, each with a ratio of 0.5. This process improved the resilience of the models to diverse image representations (more detailed information is presented in Appendix A). The assessment metrics, namely, precision, recall, F1 score, and mAP, played crucial roles in analyzing the performance of the YOLOv5 and YOLOv8 models (Appendix A). Table 1 presents a summary of the training results, specifically focusing on the variations in precision, recall, F1 score, and mAP50 for two types of images: WLI-RGB and HSI-NBI. The analysis was conducted for two detection categories: dysplasia and SCC. Our investigation showed that the YOLOv5 model, when equipped with HSI-NBI, exhibited significant superiority. The achieved precision was 77.1%, which was somewhat greater than that of the WLI model, which was 76.4%. The HSI-NBI model demonstrated exceptional performance in detecting SCC, with a remarkable precision of 85.1%. This finding was a considerable improvement of 13.9% compared with that of the WLI model. Although the WLI model exhibited a slightly greater precision of 70.2% in detecting dysplasia than the 66.4% for HSI-NBI, the overall performance of HSI-NBI was more well-rounded, especially in important categories. The confusion matrix, displayed in Table 2, provides additional evidence of the superior diagnostic skills of the HSI-NBI model, demonstrating reduced rates of false positive results. The YOLOv8 model demonstrated consistent results, confirming that the HSI-NBI model surpassed the WLI model in terms of performance, reaching superior precision rates in all aspects. The combination of the HSI-NBI method and the YOLOv5 model proved to be the most efficient approach, constantly achieving excellent results in terms of precision, recall, F1 score, mAP50, and the confusion matrix. The significant enhancements in accuracy, particularly in detecting SCC, closely corresponded with the primary goal of our work to improve the identification of early-stage EC through the utilization of modern imaging and AI technologies. Ensuring this alignment was crucial because it directly contributed to the possibility of reducing mortality rates linked to the identification of advanced-stage cancer.

A comprehensive statistical analysis was conducted using paired t-tests to rigorously evaluate and compare the performances of the YOLOv5 and YOLOv8 models across multiple metrics. This analysis specifically examined the metrics of precision, recall, F1-score, and mAP50 for two different image modalities: WLI RGB and HSI Spectrum. The objective was to ascertain whether the performance metrics had a statistically significant difference between the two models. The precision scores for the WLI RGB images demonstrated a statistically significant distinction between YOLOv5 (76.4%, 70.2%, 71.2%) and YOLOv8 (74.0%, 66.5%, 69.5%). The t-statistic of 4.44 and *p*-value of 0.047 indicated that YOLOv5 generally exhibited superior precision performance. The observed differences did not reach statistical significance, as indicated by a t-statistic of 0.83 and a *p*-value of 0.494. Similarly, no significant differences existed in the F1-scores, as indicated by a t-statistic of 2.64 and a *p*-value of 0.118. The mAP50 did not show any statistically significant difference, as indicated by a t-statistic of 1.26 and a *p*-value of 0.335. The precision differences in the HSI Spectrum images were not statistically significant, as indicated by a t-statistic of 0.117 and a *p*-value of 0.918. An evident and noteworthy pattern of significance was observed in the disparities in recall, as indicated by a t-statistic of 3.73 and a *p*-value of 0.065. This finding suggested the presence of potentially meaningful distinctions that were undetected at the conventional significance level of 0.05. The F1-scores also showed a tendency toward significance, with a t-statistic of 2.78 and a *p*-value of 0.108. Thus, YOLOv5 may have better performance in detecting more intricate details. mAP50 exhibited a nearly significant distinction, as indicated by a t-statistic of 4.29 and a *p*-value of 0.050. This finding suggested that YOLOv5 may have superior overall detection performance in the HSI modality. These statistical insights offer a numerical foundation for evaluating the comparative advantages and disadvantages of each model in various imaging scenarios. The notable improvements in precision for WLI RGB and the nearly significant improvements in recall and mAP50 for HSI indicated specific areas where YOLOv5 performed better than YOLOv8. These findings can potentially guide future enhancements and applications of these models in clinical and research settings. Comparison of the performance of the YOLOv5 and YOLOv8 models revealed that YOLOv5 had superior precision, especially in WLI RGB, where it outperformed YOLOv8 by a significant margin. Additionally, it demonstrated a slightly improved recall and mAP50 in the HSI spectrum, indicating an enhanced ability to detect objects in more intricate imaging scenarios. Although YOLOv8 had slightly lower precision and recall, it still achieved competitive mAP50 values in WLI RGB, suggesting that it can generalize well under standard imaging conditions. Nevertheless, in the HSI Spectrum, its performance slightly fell behind YOLOv5, particularly in accurate condition-specific detection tasks such as SCC. The use of the HSI spectrum significantly improved the performance of the YOLOv5 model in detecting specific conditions like SCC. Within this particular framework, the utilization of HSI enabled YOLOv5 to attain notably superior precision and F1-scores compared with the usage of WLI RGB. HSI’s capacity to capture a wider range of light and offer more intricate data about tissue characteristics greatly enhanced the model’s capacity to differentiate and precisely identify the attributes of SCC. HSI may be especially valuable for applications that demand high diagnostic accuracy in identifying and characterizing intricate medical conditions.

## 4. Discussion

EC often cannot be discovered in time because of its asymptomatic or symptom-atypical characteristics in early stage, and the early detection of EC can improve the survival rate [50,51,52]. This study underscored the potential of HSI-based conversion algorithms, showcasing their capability to convert WLI images into HSI images for effective early EC classification and detection. Despite the close similarity in overall results, this research significantly validated the accuracy of SCC predictions. Detecting dysplasia, which often appears in early phases and poses challenges for manual interpretation by pathologists, was achieved at a rate above 65%. Given the relatively low quantity of dysplasia elements in our dataset, future improvements in accuracy are anticipated with increased dataset size. This work clearly indicated that by using an HSI method with the YOLOv5 framework, the early detection of EC can be increased significantly. YOLOv5 performed better in detection compared with YOLOv8. YOLOv5 and YOLOv8 are both iterations of the YOLO object detection framework. YOLOv5 is known for its streamlined architecture, offering improved speed and accuracy in real-time object detection tasks. It excels in bounding box precision, accurately localizing objects within images. Meanwhile, YOLOv8 introduces advancements in model depth and complexity, enhancing its ability to capture intricate features. Both versions share the advantages of efficient multi-object detection, high-speed processing, and suitability for real-world applications owing to their balance between accuracy and computational efficiency. One reason that the results were not considered very good is the limited amount of training data available, coupled with existing class imbalances, as seen in the confusion matrix results in Table 2. The overall outcome may be affected by these factors, suggesting a potential for improvement by addressing class imbalances during training. Currently, the application of the SAVE method in clinical diagnoses for EC images is yet to be implemented. However, in future research on EC image processing through the SAVE method, aligning image characteristics and integrating parallel feature selection in conjunction with the detection and diagnostic techniques presented in this study can offer valuable insights for early EC diagnosis. Such an approach has the potential to enhance prediction accuracy and improve diagnostic outcomes in the future. Traditional methods of identifying EC, such as WLI and biopsy, are intrusive and may fail to detect early malignant alterations. CT and MRI scans offer anatomical overviews but do not have the necessary level of detail to detect tiny mucosal abnormalities. HSI combined with the YOLOv5 model can achieve enhanced accuracy in diagnosing early-stage EC. This integration allows one to uncover spectral information that is not visible in normal imaging techniques. Despite the need for specialized equipment and experience, this method offers fast, automated analysis and improved diagnostic accuracy, thereby boosting doctors’ capacity to detect early cancer and enhance patient outcomes.

## 5. Conclusions

The early detection of EC is crucial to enhancing patient outcomes. This work aimed to improve the early diagnosis of cancer by combining HSI with the YOLOv5 and YOLOv8 deep learning model. This approach outperformed conventional imaging approaches in terms of effectiveness and precision. Our results demonstrate that the AI-enhanced HSI method greatly enhances the accuracy of diagnosis, obtaining a precision rate of 85.1% in detecting SCC. This feature is a significant improvement compared with the YOLOv8 model. This technological innovation provides healthcare workers with more precise and practical information. It also holds the potential to enhance patient care and operational efficiency in clinical settings. Using the SAVE technique, we converted regular WLIs into high-quality NBIs, greatly enhancing our model’s ability to recognize objects. The exceptional efficacy of YOLOv5, in conjunction with HSI-NBI, showcases the revolutionary capability of cutting-edge imaging technologies in cancer diagnosis. This study ultimately filled important voids in current diagnostic methods by offering a more efficient tool for the early diagnosis of diseases. Timely and successful therapeutic interventions, with the goal of reducing mortality rates linked to late-stage cancer identification, necessitate the incorporation of advanced imaging and deep learning technology. Through the exploration of diagnostic capacities, this research improved existing approaches and laid the foundation for future advancements in medical imaging.

## Figures and Tables

**Figure 1 diagnostics-14-01129-f001:**
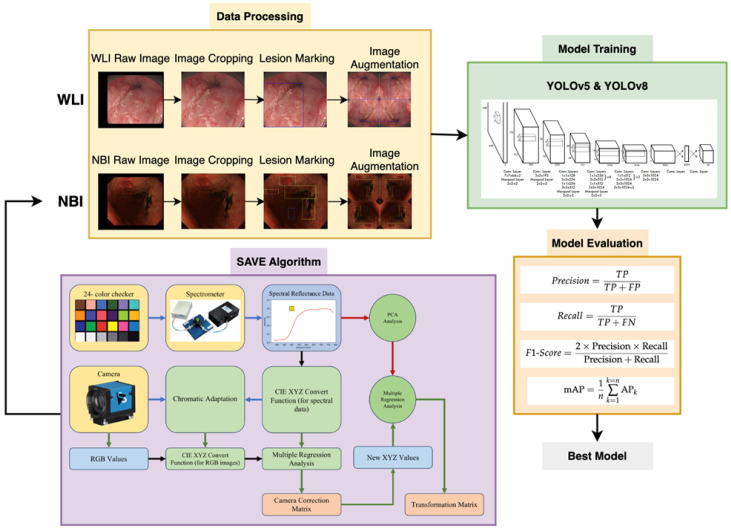
Overall flowchart of this project.

**Figure 2 diagnostics-14-01129-f002:**
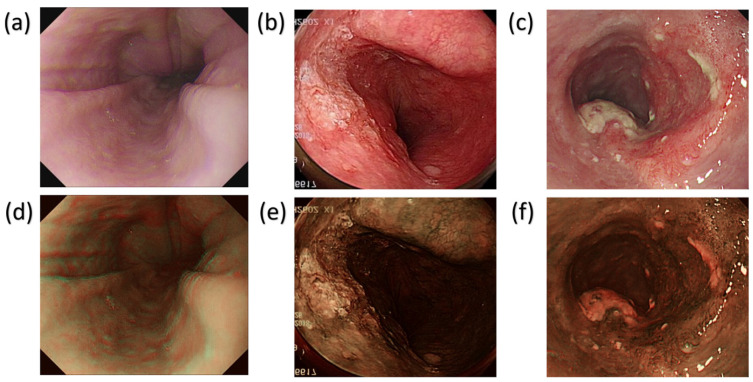
Lesion location (**a**). Normal esophageal images in WLI (**b**). Dysplasia findings in WLI (**c**). SCC findings in WLI (**d**). Normal esophageal images in HSI (**e**). Dysplasia findings in HSI (**f**). SCC findings in HSI.

**Figure 3 diagnostics-14-01129-f003:**
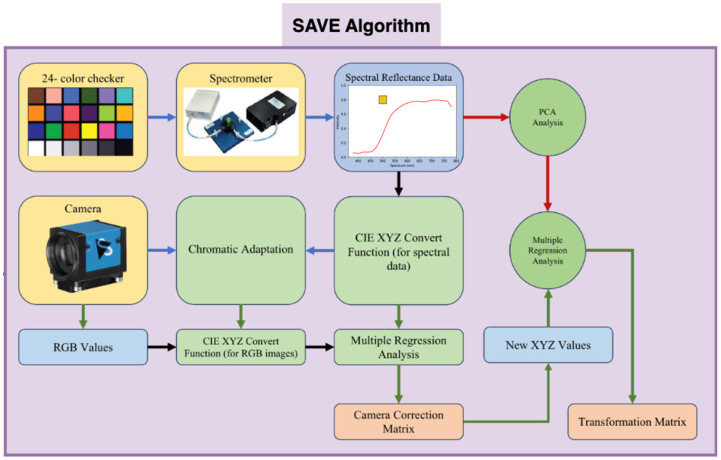
Schematics of the developed SAVE algorithm.

**Table 1 diagnostics-14-01129-t001:** Performance results analysis for training.

Framework	Model		Metrics		
YOLOv5	WLI RGB	Precision	Recall	F1-Score	mAP50
	All	76.4%	64.9%	70.2%	68.3%
	Dysplasia	70.2%	42%	52.6%	48.2%
	SCC	71.2%	69.7%	70.4%	73.6%
	HSI Spectrum	Precision	Recall	F1-Score	mAP50
	All	77.1%	63.4%	69.6%	67.5%
	Dysplasia	66.4%	43.4%	52.5%	44.8%
	SCC	85.1%	69%	76.2%	75.1%
YOLOv8	WLI RGB	Precision	Recall	F1-Score	mAP50
	All	74%	65.1%	69.3%	68.2%
	Dysplasia	66.5%	40.3%	50.2%	44.5%
	SCC	69.5%	69.7%	69.6%	73.1%
	HSI Spectrum	Precision	Recall	F1-Score	mAP50
	All	74.8%	61.4%	67.4%	65.6%
	Dysplasia	71.2%	39.8%	51.1%	43.3%
	SCC	81.7%	63.6%	71.5%	71.9%

**Table 2 diagnostics-14-01129-t002:** Confusion matrix results analysis for training.

Framework	Model		True Value			Total
YOLOv5	WLI RGB		Normal	Dysplasia	SCC	1251
	Predicted Value	Normal	1118	0	0	
		Dysplasia	3	102	1	
		SCC	2	0	25	
	HSI Spectrum		Normal	Dysplasia	SCC	1165
	Predicted Value	Normal	1031	2	1	
		Dysplasia	1	104	1	
		SCC	1	0	24	
YOLOv8	WLI RGB		Normal	Dysplasia	SCC	1253
	Predicted Value	Normal	1067	1	0	
		Dysplasia	49	96	2	
		SCC	14	0	24	
	HSI Spectrum		Normal	Dysplasia	SCC	1166
	Predicted Value	Normal	1008	1	0	
		Dysplasia	36	94	0	
		SCC	6	0	22	

## Data Availability

The data presented in this study are available upon considerable request to the corresponding author (H.-C.W.).

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
