# Peer review of "Evaluation of Spectrum-Aided Visual Enhancer (SAVE) in Esophageal Cancer Detection Using YOLO Frameworks"

_diagnostics, 2024, doi:10.3390/diagnostics14111129_

Round 1

Reviewer 1 Report

Comments and Suggestions for Authors

Dear Authors,

regarding your respectable article:

Manuscript ID: diagnostics-2964049

Article Type: Article

Paper Title: “Enhancing Early Esophageal Cancer Detection: Integration of Hyperspectral Imaging and YOLO Frameworks"

Journal: Diagnostics

The article is presenting one of the early detection of esophageal malignancy is crucial for improving patient outcomes, as the disease often presents at an advanced stage with poor prognosis. Esophageal cancer has a high mortality rate due to late diagnosis, highlighting the critical importance of early detection strategies. Thus, the early detection and effective therapy is necessary for appropriate treatment strategies and improving patient outcomes.

I have gone thru the whole article; the article`s idea is interested and comes under the scope of the Journal. The authors report about the early detection for the “esophageal malignancy” thru investigating You Only Look Once (YOLO) frameworks, which considered to be a type of deep learning architecture designed for real-time object detection, offering a fast and efficient approach compared to the conventional methods. The authors did a great job to write and elaborate the article. However, the report itself suffers from an overly opaque parts, simplistic design, unusual diction, and poor grammar. But it could be more appropriate after a major correction before re-submission as well as need to eliminate the opaque and unclear parts. Where, the following aspects of the paper need to be improved:

1.     The current title of the article is too long and may seems to be unattractive which needs to be modified and revised by the authors. Kindly make it shorter, concise, avoid any un-necessary words, and make it in brief informative for the presented study, to grab the reader`s attention with attractive expressions and avoid unnecessary abbreviations.

2.     The language of the whole paper not bad but still need to be checked since it makes reading and understanding somehow difficult and there are some of incomplete sentences and overuse of conjunctions. It could be revised through language program or by native colleague.

3.     The “Abstract” section it needs to be revised by the authors and re-editing in a better presentable manner without skipping the ideas to comply with the presented idea of the manuscript and more organized.

4.     Kindly in the “Abstract & Introduction” section try to elaborate the study motivation and the main problem, then start to illustrate your study`s objective with your contribution to overcome the conventional methods or the improvements.

5.     The authors should highlight the motivation for this study with a clear idea or aim. Additionally, addressing the importance of the research with objectives and clear justification for their hypothesis.

6.     The “Introduction” section needs to be revised by the authors and re-edit in a better order with more exploration of the previous recent studies in the same field and highlighting the new and recent studies with its advantages and limitations, especially for the HSI. kindly read some of these useful researches which could be helpful in your research and self-knowledge about the HSI applications toward the cancer detection:

*       "Hyperspectral imaging for clinical applications." BioChip Journal 16.1 (2022): 1-12.

*       "Applications of hyperspectral imaging in the detection and diagnosis of solid tumors." Translational Cancer Research 9.2 (2020): 1265.

*       "Emerging technology for intraoperative margin assessment and post-operative tissue diagnosis for breast-conserving surgery." Photodiagnosis and Photodynamic Therapy 42 (2023): 103507.

*        "Hyperspectral imaging: a review and trends towards medical imaging." Current medical imaging 19.5 (2023): 417-427.

*       "Recent advances of hyperspectral imaging application in biomedicine." Chinese Journal of Lasers 45.2 (2018): 0207017. 

7.     The literature review is not complete to cover both the previous research studies and how the technique (YOLOv5 and YOLOv8) could be identified and compared with the conventional methods. Additionally, write a brief comparison about the conventional therapy techniques vs your proposed technique (HSI and its various applications in the cancer detection) to elaborate its advantages and disadvantages.

8.     The investigated imaging techniques (Hyperspectral imaging / conventional endoscope) is quite different in its output (cube image / pixel color image), how you compromise your data bank in your investigation to using both of these two totally different outcomes?

9.     The data bank for the patient s should be more clarified and attached in the article appendix for better clarification for the researchers.

10.  The innovation of this paper is limited and not clear if the proposed AI new technique for THE early detection is reliable and better than the conventional methods, especially regarding the medical sector and the surgeons later, please illustrate in more details research contribution and how the proposed technique could be helpful in the medical sector or even how it could assist the medical stuff.

11.  The “Statistical analysis” section is not clear how it improves the investigation`s outcome, please try to clarify and revise this section with its results.

12.  Kindly revise the “Discussion section” to cover the study`s objective and linking it with their study`s outcome. This section should not contain any comments or interpretations of the experimental results obtained.

13.  kindly read some of these useful researches which could be helpful in your research outcome and could improve your “discussion section”, for example:

*        "Recent advances in early esophageal cancer: diagnosis and treatment based on endoscopy." Postgraduate Medicine 133.6 (2021): 665-673.

*        "Staging early esophageal Cancer." Stem Cells, Pre-neoplasia, and Early Cancer of the Upper Gastrointestinal Tract (2016): 161-181.

*         "Current trends in endoscopic diagnosis and treatment of early esophageal cancer." Cancers 13.4 (2021): 752.

14.  The “conclusion section” needs to be revised to elaborate the article`s motivation and its investigation outcome, try to revise and re-edit in a better presentable way.

15.  Kindly revise the reference section and update it, to cover your study`s background and literature review. Additionally, to be in the latest and nearest year to your research, as some reference needs to be replaced as it is too old regarding your proposed study, such as: (Ref #3-2013 / Ref #6-2010 / Ref #8-2015 / Ref #11-2011 / Ref #12-2014 / …etc.).

 Finally, I really appreciate all the hard work and efforts done by the Author and his team. However, I think they needs an additional effort in this manuscript to re-organize the ideas and highlight the investigation`s outcome with more details and clear illustration. I also recommend to revise the article with one of the scientific program language or a foreign expert colleague to be more reliable and improve the language speech output.

Comments on the Quality of English Language

Dear Authors,

I recommend to revise the article with one of the scientific program language or a foreign expert colleague to be more reliable and improve the language speech output.

Best regards and good luck,

Reviewer 2 Report

Comments and Suggestions for Authors

I have the following comments:

1) Abstract (lines 28-30). The sentence: 'The dataset ... robust detection' could be removed for the sake of conciseness, and the case distribution among training, validation and test sets can be briefly reported later on in the abstract, e.g. in the following sentence.

2) Abstract (lines 38-39 and 41-44). Please provide quantitative findings related to the use of inferential statistical methods (e.g., CI95 confidence intervals and p-values) for the comparisons of the various performance scores, i.e. precision rates and F1 scores.

3) Introduction (lines 60-61). In addition to endoscopy (which is the mainstay for local staging of esophageal cancer), I suggest adding a brief description of the role that radiology (transesophageal ultrasonography, CT, MRI) can play in the diagnostic management, how it can complement endoscopy, its strengths and limitations.

4) Introduction (lines 89-123). This large part of the Introduction illustrates the main findings from several literature articles. I suggest that it be drastically reduced (by more than 50%), summarizing the key findings from at most 2-3 articles and highlighting which gaps in current knowledge could be filled by the present study. At least part of the remaining articles could be commented in the Discussion section and compared with the findings of the present study.

5) Results. As anticipated in comment #2, it is highly recommended to systematically compare the performance of the YOLOv5 and YOLOv8 models using inferential statistical analysis - this would add much scientific rigor and value to the study. Such methods should be illustrated in an additional subsection (2.6) entitled: 'Statistical analysis'.

6) Some minor English language editing should be performed throughout the manuscript. At line 168, please replace 'symptoms' (a clinical term) with 'findings'.

Comments on the Quality of English Language

Some minor English language editing should be performed.

Round 2

Reviewer 1 Report

Comments and Suggestions for Authors

Dear Authors,

I really appreciate all the hard work done in your updated version. However, I still have some concerns, such as the following:

1- please identify and more clarify your "Method and material" section to elaborate your work.

2- kindly revise your "Results" section and linking it with your research objective.

3- kindly revise your "Conclusion" section, it could be better.

4- kindly revise your "Reference" section and avoid using unnecessary old references such as in Ref#8-1983 ,Ref#10-2011....ect.

good luck and best regards,

Comments on the Quality of English Language

Dear Sir,

I appreciate the diligent efforts undertaken in the revised version. However, I note that the language remains unedited. Consider seeking assistance from a colleague or utilizing an AI program to enhance the quality of your article.

Best regards,

Reviewer 2 Report

Comments and Suggestions for Authors

Thank you. No further comments.

Author Response

1) Thank you. No further comments.

Reply:

        Thank you very much for your feedback and for reviewing our manuscript. We are delighted to hear that there are no further comments at this stage. Your insights have been invaluable to the refinement of our research. We appreciate your time and effort in helping us improve our work.